# Coordination of agricultural informatization and agricultural economy development: A panel data analysis from Shandong Province, China

Yu Sun[ID], Zhe Zhao, Mingquan Li[ID]*

College of Economics and Management (Cooperative), Qingdao Agricultural University, Qingdao, China

* sylyac2006@126.com

## Abstract

With the continuous development of the world economy, science, and technology, the era of intelligence and information is upon us. Through the implementation of the digital rural construction project in China, agriculture is developing rapidly in the direction of informatization. As a major agricultural province in China, Shandong Province has been ranked first in China for many years in terms of gross agricultural product and the import and export of agricultural products. According to the current situation of agricultural informatization in Shandong Province, this study uses relevant evaluation index data of Shandong Province from 2011 to 2019 and applies the entropy value method, coupling degree analysis model, and coupling coordination degree analysis model to deeply evaluate the mutual influence and coordination degree between the agricultural informatization level and agricultural economic development in Shandong province in each year. We identify the possible problems in the development of agricultural informatization in Shandong province. Finally, in terms of talent construction, infrastructure construction, the main role of government, policies, and farmers' awareness of informatization, we propose some countermeasures and suggestions that are conducive to the coordinated development of agricultural informatization and the agricultural economy.

## Introduction

With the continuous development of the world economy and science and technology [1, 2], the era of intelligence and information is upon us [3, 4]. As one of the largest emerging countries in the world, China has made progress in many areas [5, 6]. At the same time, China is also a largely agricultural country with a large population [6, 7]. Understanding how to obtain crop growth information in a timely and accurate manner and improve crop production efficiency has become a hot topic in agricultural development [8, 9]. Modern information technology is unimpeded and developed, providing an excellent development channel for agriculture [10, 11]. To develop agriculture and increase its output value [12], it is necessary to

**Funding:** Research and planning fund for Humanities and social sciences of the Ministry of Education: Research on the motivation, behavior decision-making and incentive mechanism of small farmers adopting eco agricultural technology under the background of food security strategy; Shandong humanities and Social Sciences application research project: the theoretical framework, efficiency increasing mechanism and realization path of digital economy enabling the high-quality development of agriculture in Shandong Province (2022-YYJJ-12). This research was funded by Shandong Province Modern Agricultural Industrial Technology System (SDAIT-01-13); Research and planning fund for Humanities and social sciences of the Ministry of Education: Research on the motivation, behavior decision-making and incentive mechanism of small farmers adopting eco agricultural technology under the background of food security strategy; Shandong humanities and Social Sciences application research project: the theoretical framework, efficiency increasing mechanism and realization path of digital economy enabling the high-quality development of agriculture in Shandong Province (2022-YYJJ-12); the Project of Shandong Provincial Department of Education (M2020111); Qingdao Social Science Planning Project: Research on the Interactive Relationship Between Agricultural Informatization and Agricultural Economic Growth in Qingdao Under the Background of Digital Rural Strategy (QDSKL2101221); and Qingdao Double Hundred Research Project: Research on the Coupling Mechanism and Guiding Policies of Digital Investment and Agricultural Modernization in Qingdao (2021-B-15); Soft Science Project of Science and Technology Department of Shandong Province(2021RKY07139). The funders had no role in study design, data collection and analysis, decision to publish, or preparation of the manuscript.

**Competing interests:** The authors have declared that no competing interests exist.

speed up the pace of agricultural informatization and realize the effective combination of agricultural production and information technology [13, 14].

China's 14th Five-Year Plan specifies that it is important to accelerate agricultural modernization and realize rural revitalization to vigorously promote the deep integration of the new generation of information technology and the agricultural real economy based on the new generation of information technology, such as smart agriculture, artificial intelligence, Internet, and big data [15, 16]. According to the No. 1 Document of the central government in 2020, a big data management center for agriculture and rural areas should be built to accelerate the application of such new information technologies. In 2021, the No. 1 Central Document explicitly proposed comprehensively promoting rural revitalization, implementing the digital rural construction and development project, and comprehensively promoting the rural agricultural modernization plan [17, 18]. The implementation of several documents has shown that the Chinese government attaches great importance to the construction of agricultural informatization [11, 19]. Since the 21st century, the history teaching experience of the development of agriculture-related enterprises has shown that the level of agricultural informatization is the foundation for the continuous development of the agricultural economy, and the agricultural and rural economy also drives the construction of agricultural informatization in the development [20, 21]. Because the level of agricultural informatization is closely related to the development of the agricultural economy [22, 23], this paper deeply examines the coordination of agricultural informatization and agricultural economic development in Shandong Province.

Many scholars have studied the relationship between agricultural informatization and the agricultural economy. In terms of the research on the connotation of agricultural informatization [24], Porat [25] studied the connotation and characteristics of agricultural informatization and conducted theoretical discussions based on the practice of agricultural informatization and the promotion of economic development, agricultural informatization, and agricultural sustainable development. Accelerating agricultural informatization is the inevitable choice of agricultural sustainable development [26]. Some scholars [16, 27] believed that agricultural informatization should be understood as a broad concept that encompasses the whole process of agriculture and relies on modern information technology equipment, such as information networks and digitization, to support agricultural operation, monitor and manage agricultural resources, support agricultural economic development, and promote rural social informatization. Other scholars [28, 29] established a relevant index system from the aspects of economic benefits, social benefits, and ecological and environmental benefits to comprehensively evaluate the benefits of agricultural informatization for understanding the connotation of agricultural informatization.

Studies on the role of agricultural informatization are mainly reflected in the contribution of agricultural informatization to the agricultural economy [30], the integration of agricultural informatization and agricultural modernization [31, 32], and the relationship between agricultural informatization and the rural economy [33, 34]. Some scholars have studied the contribution of agricultural informatization to the agricultural economy and have put forward countermeasures and suggestions to improve the level of informatization [35, 36]. Other scholars also have constructed an indicator system of agricultural informatization and agricultural modernization, evaluated the integration degree of agricultural informatization and agricultural modernization in the country and major regions, and put forward countermeasures and suggestions for the integration of agricultural informatization and agricultural modernization [37, 38].

In summary, scholars' research on the relationship between agricultural informatization and agricultural economy has generally established that the two have irreplaceable roles of mutual promotion and integration, but few reports have been published on the coordination of the two, especially quantitative regional research.

Therefore, the research purpose of this paper is to (1) take Shandong Province of China as an example on the basis of previous studies, use the entropy value method and coupling coordination analysis, and adopt relevant data on Shandong Province from 2011 to 2019 to build the coupling degree evaluation and coordination degree evaluation models; and (2) calculate the coupling degree and coordination degree of annual agricultural informatization and agricultural economic development in Shandong Province, comprehensively analyze the coordination degree between them, and propose relevant countermeasures and suggestions.

## Overview of the study area

### Improvement of the agricultural intelligence level

As the new generation of information technologies (e.g., big data and the Internet of Things, IoT) continues to be integrated into the agricultural sector, the transformation of agricultural, scientific, and technological achievements has been significantly accelerated, and the production process of agriculture has been continuously optimized [11, 39]. In Shandong Province, livestream e-commerce of agricultural products has evolved rapidly. In livestream selling, Shandong Province has also been constantly optimizing the circulation of agricultural products. Agricultural information technologies, such as agricultural database system, agricultural decision-making system, management information system, 3S technology, expert system, global positioning system, computer network, agricultural multimedia technology, and remote communication, have emerged successively [40, 41].

### Various ways for farmers to obtain information

The more information resources that farmers get, the more beneficial it will be to carry out agricultural information construction, which can promote the rapid development of the agricultural economy [42, 43]. With the rapid development of agricultural informatization in Shandong Province, the way for farmers to obtain agricultural information has become more flexible and efficient. Information transmission is no longer in the form of the original communication between people in the past; it takes place through computers and modern communication networks to spread news. Such a change has greatly improved the efficiency and quality of farmers' access to agricultural information [16, 18].

### Improvement of the agricultural information service system

With the rapid development of the new generation of information technology, the integration of information technology in the agricultural field has become more in-depth, and the agricultural information service system in Shandong Province also has become more efficient [44, 45], which is embodied in several qualities. First, the agricultural information service network platform has improved. Among these improvements, the Shandong Province rural agricultural informatization integrated service platform launched rural agricultural IoT online interactive features, services, and intelligent equipment service support, agricultural e-commerce, a data resource integration system, an intelligent agricultural system, and remote video interactive features. These features have allowed the agricultural information service platform to better fuel the agricultural economic growth of the service function. Second, based on the provincial service platform, Shandong Province has built its own agricultural information service platform in cities and counties and gradually has improved the agricultural information network service system so that people in cities and counties can access a local service website to understand the local agricultural information status efficiently and succinctly.

### Improvement of agricultural informatization infrastructure

The improvement of infrastructure is conducive to the improvement of information occlusion and information asymmetry in rural areas and is a necessary preparation for the construction of agricultural informatization [46, 47]. At present, Shandong Province has radio, television, mobile phone, broadband, and mobile Internet rapid development. As shown in Table 1, in 2019, the total business volume of posts and telecommunications in Shandong Province reached 6504.6 billion yuan and 578.661 billion yuan, respectively. At the same time, the number of mobile Internet users reached 88.553 million, and the number of Internet broadband access users exceeded 30 million, reaching the level of 31.861 million. In addition, the radio and television industry has developed rapidly. By 2019, the coverage rate of both radio and television in Shandong province exceeded 98%. This is conducive to farmers gaining faster and more convenient access to more agricultural information to promote the development of agricultural informatization.

## Construction of an evaluation index system based on coordination analysis

### Selection of an evaluation index system

This research is not a separate evaluation of agricultural information and agricultural economic development level, but rather is a qualitative analysis of Shandong Province's agricultural informatization level and agricultural economic development level based on building the model of quantitative analysis of the coordination degree between them. Therefore, when choosing the harmony of the two indexes, the connotation of the comprehensive considers the index data, availability, and representativeness, based on the research results of Li [13], Akbar [38], and Li [42], and an indicator system for measuring the coordination between agricultural informatization and agricultural economic development level in Shandong Province is determined, as shown in Table 2.

### Data sources and data optimization

The data used in this paper are from the Statistical Yearbook of Shandong Province, the Statistical Yearbook of China, and the annual statistical bulletin published on the website of the

**Table 1. Development status of agricultural informatization infrastructure in Shandong Province.**

| | Railway Mileage (km) | Mileage of Highway open to traffic (km) | Total business volume of posts and telecommunications (100 million yuan) | Total Telecom Business (100 million yuan) | Internet Users (10,000 households) | Broadband Access Users (10,000 households) | Broadcast Coverage (%) | TV Coverage (%) |
|---|---|---|---|---|---|---|---|---|
| 2011 | 4177.0 | 233189.0 | 771.2 | 723.6 | 4371.0 | 1154.1 | 98.2 | 97.9 |
| 2012 | 4306.0 | 244586.0 | 849.2 | 797.6 | 4865.1 | 1364.1 | 98.3 | 98.0 |
| 2013 | 4397.0 | 252785.0 | 919.7 | 863.7 | 5556.1 | 1465.1 | 98.5 | 98.2 |
| 2014 | 4546.0 | 259514.0 | 1213.6 | 1067.9 | 5569.1 | 1523.9 | 98.7 | 98.5 |
| 2015 | 4863.0 | 263447.0 | 1458.6 | 1253.1 | 6109.6 | 1625.7 | 98.8 | 98.6 |
| 2016 | 4882.0 | 265720.0 | 1165.0 | 863.4 | 7391.2 | 2366.5 | 99.0 | 98.6 |
| 2017 | 5115.0 | 270590.0 | 1887.7 | 1494.8 | 8508.0 | 2588.7 | 99.1 | 98.9 |
| 2018 | 5676.0 | 275642.0 | 4180.3 | 3651.9 | 9552.3 | 2884.8 | 99.1 | 99.1 |
| 2019 | 5972.0 | 280325.0 | 6504.6 | 5786.6 | 8855.3 | 3186.1 | 99.1 | 99.1 |

Data source: Statistical Yearbook of Shandong Province (http://tjj.shandong.gov.cn/col/col6279/index.html).

**Table 2. Evaluation index system.**

| Level indicators | The secondary indicators | Unit |
|---|---|---|
| Level of agricultural informatization | Total post and telecommunications business X1 | One hundred million yuan |
| | Total Telecom Business volume X2 | One hundred million yuan |
| | Mobile Internet users X3 | Thousands of families |
| | Broadband access user X4 | Thousands of families |
| | Rural electricity consumption X5 | Billion kwh |
| | Rural telephone users X6 | Thousands of families |
| | Number of computers used per 100 people X7 | Tai |
| Level of agricultural economy | Gross agricultural output value X8 | One hundred million yuan |
| | Per capita net income of farmers X9 | yuan |

Data source: China Statistical Yearbook and Shandong Statistical Yearbook.

Shandong Provincial government. The index data of Shandong Province in the past nine years from 2011 to 2019 are obtained, as shown in Table 3 and Fig 1.

Because of the different characteristics and properties of each evaluation index in the evaluation index system, the direct use of original data for research and analysis may exaggerate the impact of the evaluation index with a high value in the analysis and may reduce the role of the evaluation index with a small value in the analysis [48]. Therefore, to ensure the reliability of the calculation results, the data in Table 3 are normalized according to the min-max standardization method in this paper. The specific formula is as follows:

$$Y_{ij} = (X_{ij} - X_j \min)/(X_j \max - X_j \min), \tag{1}$$

where $Y_{ij}$ represents the normalized value of the jth (j = 1,2,3,4,5,6,7,8,9) index in the ith (i = 1,2,3,4,5,6,7,8,9) year; $X_{ij}$ represents the original data of the jth index in the ith (i = 1,2,3,4,5,6,7,8,9) year; $X_j \min$ represents the minimum original data of each evaluation index in nine years; and $X_j \max$ represents the largest original data in nine years under each evaluation index.

According to formula (1), the original data in Table 3 can be standardized, and the calculation results are shown in Table 4.

**Table 3. Indicator data of Shandong Province from 2011 to 2019.**

| Years | X1 | X2 | X3 | X4 | X5 | X6 | X7 | X8 | X9 |
|---|---|---|---|---|---|---|---|---|---|
| 2011 | 771.2 | 723.6 | 4371.0 | 1154.1 | 456.5 | 809.0 | 7.8 | 3737.0 | 8342.1 |
| 2012 | 849.2 | 797.6 | 4865.1 | 1364.1 | 465.8 | 786.8 | 8.0 | 3829.2 | 9446.5 |
| 2013 | 919.7 | 863.7 | 5556.1 | 1465.1 | 471.4 | 712.2 | 15.8 | 4335.8 | 10686.9 |
| 2014 | 1213.6 | 1068.9 | 5569.2 | 1523.9 | 480.0 | 538.9 | 16.9 | 4556.1 | 11882.3 |
| 2015 | 1458.6 | 1253.1 | 6109.6 | 1625.7 | 482.3 | 343.9 | 18.0 | 4662.6 | 12930.4 |
| 2016 | 1165.0 | 863.4 | 7391.2 | 2366.5 | 488.8 | 292.2 | 18.0 | 4387.5 | 13954.1 |
| 2017 | 1887.7 | 1494.8 | 8508.0 | 2588.7 | 488.5 | 245.0 | 20.0 | 4403.2 | 15117.5 |
| 2018 | 4180.3 | 3651.9 | 8552.3 | 2884.8 | 416.2 | 228.3 | 24.0 | 4678.3 | 16297.0 |
| 2019 | 6504.6 | 5786.6 | 8855.3 | 3186.1 | 435.9 | 235.5 | 27.0 | 4914.4 | 17775.5 |

Data source: China Statistical Yearbook (http://www.stats.gov.cn/tjsj/ndsj/) and Shandong Statistical Yearbook (http://tjj.shandong.gov.cn/col/col6279/index.html).

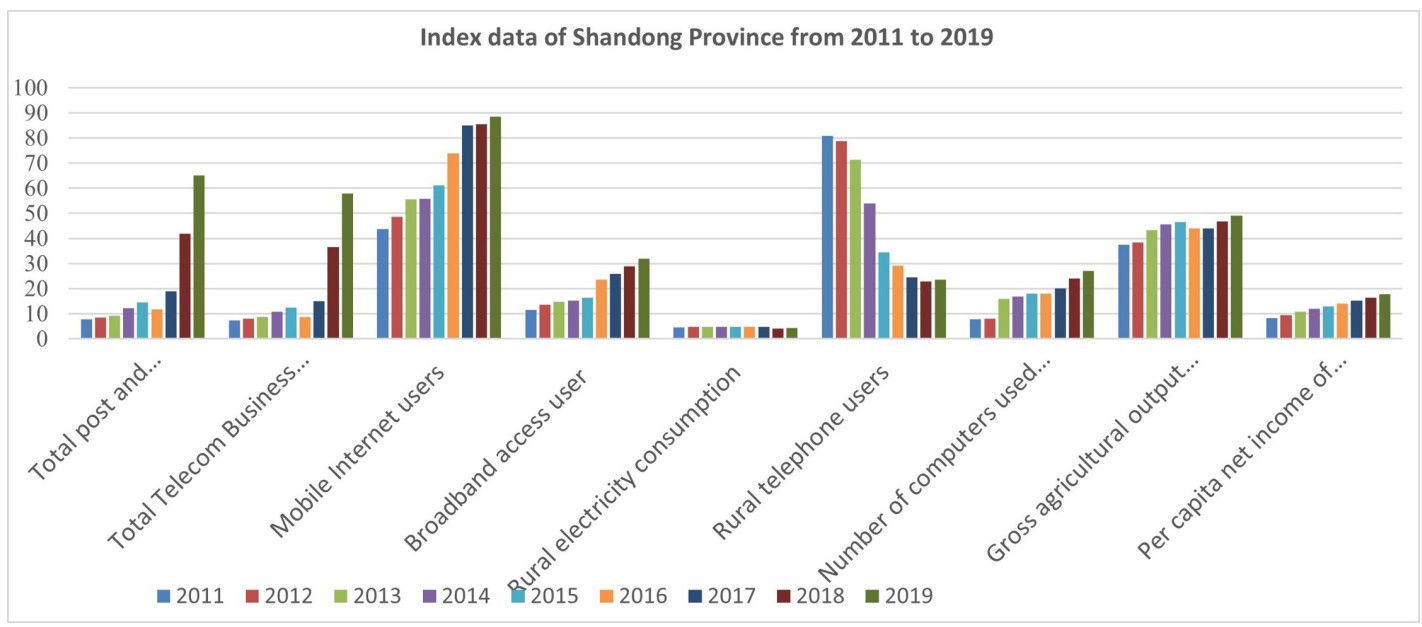

**Fig 1. Indicator data of Shandong Province from 2011 to 2019.** Data source: China Statistical Yearbook (http://www.stats.gov.cn/tjsj/ndsj/) and Shandong Statistical Yearbook (http://tjj.shandong.gov.cn/col/col6279/index.html).

## Analysis model of coordination between agricultural informatization and agricultural economic development

### Entropy value method

The entropy value method is an important application of entropy theory in the field of determining weight [49, 50]. The degree of dispersion and the importance of analysis of each evaluation index can be judged by the entropy value and the weight of the index [51, 52]. Generally, the smaller the entropy value, the greater the variation degree of the index value, the greater the weight coefficient, the more information provided, the greater the role played in the comprehensive evaluation, and vice versa [52, 53]. The rationality of the index weight coefficient directly affects the reliability of the comprehensive evaluation result and even the correctness of the decision. Entropy method is a combination of static weight and dynamic weight, and its

**Table 4. Indicator data standardization table of Shandong Province from 2011 to 2019.**

| Years | X1 | X2 | X3 | X4 | X5 | X6 | X7 | X8 | X9 |
|-------|------|------|------|------|------|------|------|------|------|
| *2011* | 0.000 | 0.000 | 0.000 | 0.000 | 0.555 | 1.000 | 0.000 | 0.000 | 0.000 |
| *2012* | 0.014 | 0.015 | 0.110 | 0.103 | 0.683 | 0.962 | 0.010 | 0.078 | 0.117 |
| *2013* | 0.026 | 0.028 | 0.264 | 0.153 | 0.760 | 0.833 | 0.417 | 0.509 | 0.249 |
| *2014* | 0.077 | 0.068 | 0.267 | 0.182 | 0.879 | 0.535 | 0.474 | 0.696 | 0.375 |
| *2015* | 0.120 | 0.105 | 0.388 | 0.232 | 0.910 | 0.199 | 0.531 | 0.786 | 0.486 |
| *2016* | 0.069 | 0.028 | 0.674 | 0.597 | 1.000 | 0.110 | 0.531 | 0.552 | 0.595 |
| *2017* | 0.195 | 0.152 | 0.923 | 0.706 | 0.995 | 0.029 | 0.635 | 0.566 | 0.718 |
| *2018* | 0.595 | 0.578 | 0.932 | 0.852 | 0.000 | 0.000 | 0.844 | 0.799 | 0.843 |
| *2019* | 1.000 | 1.000 | 1.000 | 1.000 | 0.271 | 0.012 | 1.000 | 1.000 | 1.000 |

Data source: Calculated according to Table 3.

innovation is flexible, which makes up for the shortcomings of the subjective weight method and enhances the scientific and rational evaluation. To avoid influencing the calculation result of the analysis, this paper adopts the entropy value method to determine the weight of each index [54, 55]. The specific calculation steps are as follows:

1. Calculate the proportion of the index value according to the standardized index data:

$$P_{ij} = Y_{ij} / \sum\nolimits_{(i=1)}^{n} Y_{ij}, \ (i = 1, \ 2, \ 3, \ \ldots, \ n) \tag{2}$$

where Pij represents the proportion of the j-th indicator in the i-th year.

2. Calculate the information entropy of each evaluation index (i.e., describe the uncertainty of the occurrence of each possible event of the information source):

$$E_j = -\frac{1}{\ln(n)} \left\{ \sum_{i=1}^{n} P_{ij} \ln (P_{ij}) \right\} \tag{3}$$

Among them, Ej represents the information entropy of the jth index.

3. Calculate the utility value of information:

$$Dj = 1 - Ej \tag{4}$$

4. Calculate the weight of each evaluation index:

$$Wj = Dj / (n - \sum\nolimits_{i=1}^{n} E) \ (j = 1, \ 2, \ 3, \ \ldots, \ n), \tag{5}$$

where Wj represents the weight of the jth evaluation index. As there are nine evaluation indicators, the number of years, n, is 9 here. Also, $\sum_{i=1}^{n} W = 1$.

5. U represents the comprehensive evaluation value of agricultural informatization and agricultural economic development in each year:

$$U = \sum\nolimits_{i=1}^{n} WYij \tag{6}$$

## Coupling degree evaluation model

This paper chooses the entropy value method to the agricultural informatization level and agricultural economic development level of evaluation index to objectively, then use the agricultural informationization and agricultural economic development, the coupling measure model of agricultural informationization and the paper studies the coupling degree of coordinated development of agricultural economy, using spatial statistical tools at the same time, spatial pattern analysis of the coupled coordination degree.

The coupling degree evaluation model is used to measure the interaction and influence between two or more systems to reflect the degree of interdependence and mutual restriction between systems [56, 57]. In this paper, the coupling degree model of agricultural informatization and agricultural economic development is regarded as the coupling relationship between "agricultural informatization" and "agricultural economic development" [58]. The calculation

formula follows:

$$C = \{(U1 \times U2)/(U1/2 + U2/2)2\}1/2, \tag{7}$$

where C represents the coupling degree of agricultural informatization and agricultural economic development, U1 represents the comprehensive evaluation index of agricultural informatization, and U2 represents the comprehensive evaluation index of agricultural economic development. The coupling degree C ∈ [0, 1]; when C = 1, the coupling degree reaches the maximum, indicating that the mutual influence between agricultural informatization and agricultural economic development reaches the maximum, and when C = 0, the coupling degree reaches the minimum value, indicating that there is no interaction between the agricultural informatization level and the agricultural economic development, which means that the two will develop in the direction of disorder. The larger the value of C, the stronger the coupling effect between them [52, 59].

## Coordination evaluation model

The coupling degree analysis model mainly reflects the degree of mutual promotion and mutual restriction between agricultural informatization and agricultural economic development, without distinguishing the direction of action. It is not able, however, to reflect the degree of coordination between them [57, 60]. Therefore, to reflect the level of coordination between them more clearly, this paper constructs the evaluation model of the coupling coordination degree between agricultural informatization and agricultural economic development based on the coupling degree model. The model expression is as follows:,

$$D = C \times T, \tag{8}$$

$$T = \alpha U1 \times \beta U2, \tag{9}$$

where D represents the degree of coordination between agricultural informatization and agricultural economic development, and D ∈ [0, 1]. When D = 1, the coordination degree between agricultural informatization and agricultural economic development reaches the best state, and the value of D tends toward 0, indicating that the degree of coordination between them is lower. When the value of D tends toward 1, this indicates that the degree of coordination between the two is higher. T represents the comprehensive index of agricultural informatization and agricultural economic development; U1 is the comprehensive evaluation index of agricultural informatization; U2 is the comprehensive evaluation index of agricultural economic development; and α and β are contribution coefficients of agricultural informatization and agricultural economic development, respectively. In this paper, in the study of the degree of coupling coordination between agricultural informatization and agricultural economic development, both are considered to be very important, and α + β = 1, so α = β = 0.5 is assumed [61, 62]. According to the calculated coupling coordination degree, 10 grading standards of coordination degree are determined, as shown in Table 5.

## Analysis results of coordination between agricultural informatization and agricultural economic development

### Determine the weight of evaluation indicators

The weight model of each evaluation index is obtained according to the entropy method, as shown in Table 6.

**Table 5. Classification criteria of coupling coordination degree.**

| Coordination degree D value interval | Coordination level | Coordination degree |
|---|---|---|
| (0.0–0.1) | 1 | Extreme imbalance |
| [0.1–0.2) | 2 | Serious imbalance between |
| [0.2–0.3) | 3 | Moderate disorders |
| [0.3–0.4) | 4 | Mild disorder |
| [0.4–0.5) | 5 | On the verge of disorder |
| [0.5–0.6) | 6 | Barely coordinated |
| [0.6–0.7) | 7 | Mild coordination |
| [0.7–0.8) | 8 | Moderate coordination |
| [0.8–0.9) | 9 | Good coordination |
| [0.9–1.0) | 10 | High-quality coordination |

## Weight analysis of evaluation indicators

By calculating the entropy value of the nine evaluation indexes, it can be found that the information reflected by the nine evaluation indexes is more ideal, indicating that the evaluation index system is more perfect, and it is more scientific to use the entropy value method to assign weight to the indexes. From the perspective of the weight of nine second-level indicators, the weight of the total post and telecommunications service, total telecom service, and rural telephone volume is greater than the mean weight (0.111), and the weights are 0.204, 0.227, and 0.145 respectively, with the cumulative weight reaching 0.576. This shows that the three evaluation indexes have the greatest effect on measuring the coordination degree of agricultural informatization and agricultural economic development. Because these three indicators belong to the index content of agricultural informatization, the development level of agricultural informatization is the main factor affecting the coordinated development of the two. Therefore, speeding up the construction of agricultural informatization is the highest priority among the priorities to coordinate the development of agricultural informatization and the agricultural economy.

## Coupling degree and coordination degree level

In this paper, formulas (7)–(9) are used to calculate the coupling coordination analysis results of agricultural informatization and agricultural economic development in Shandong Province

**Table 6. Determination of the weight of evaluation indexes.**

| Level indicators | Secondary indicators | Entropy E | The weight W |
|---|---|---|---|
| *Level of agricultural informatization* | Total post and telecommunications business | 0.644 | 0.204 |
| | Total telecom business volume | 0.605 | 0.227 |
| | Mobile Internet users | 0.862 | 0.079 |
| | Broadband access users | 0.832 | 0.097 |
| | Rural electricity consumption | 0.921 | 0.045 |
| | Rural telephone users | 0.747 | 0.145 |
| | Number of computers used per 100 people | 0.870 | 0.075 |
| *Level of agricultural economy* | Gross agricultural output value | 0.897 | 0.059 |
| | Per capita net income of farmers | 0.881 | 0.068 |

Data source: Weight model calculated using entropy method.

from 2011 to 2019, as shown in Table 7, and the coupling degree and coordination degree of agricultural informatization and agricultural economic development in every city of Shandong Province from 2016 to 2019, as shown in Table 8.

## Coupling degree analysis

As shown in Table 8, the coupling degree of agricultural informatization and agricultural economic development keeps rising on the whole. The coupling degree of the two increases from 0.145 in 2011 to 0.793 in 2015, then decreases to 0.66 in 2016, and continues to rise to 0.701 in 2019. On the whole, "agricultural informatization" and "agricultural economic development" are increasingly interdependent and mutually restricted, which is reflected mainly in the rapid growth of the agricultural economy and is greatly promoted by the development of agricultural informatization in Shandong Province.

For further analysis, this paper divides cities according to the coupling degree of 16 prefecture-level cities in Shandong Province:

1. Low-intensity coupling cities (0.20–0.40): From 2016 to 2019, the number of cities of this type is two, one, two, and two, and the proportion is 12.5%, 6.25%, 12.5%, 12.5%.

2. Medium-intensity coupling cities (0.40–0.75). From 2016 to 2019, the number of cities with this type is four, six, six, and two, accounting for 25%, 37.5%, 37.5%, and 12.5%, respectively.

3. High-intensity coupling cities (0.75–1). From 2016 to 2019, the number of such cities is 10, 9, 8, and 12, accounting for 62.5%, 56.25%, 50%, and 75%, respectively. The coupling degree between agricultural informatization and agricultural economic development in Shandong Province is generally high, and the city types are mainly highly coupled cities, with the proportion of highly coupled cities reaching 3/4 in 2019.

The ArcGIS platform is further used to conduct spatial visualization processing of the coupling degree between agricultural informatization and agricultural economic development in Shandong Province from 2016 to 2019, as shown in Fig 2. The coupling coefficients of Jinan (0.965), Qingdao (0.977), and Weifang (0.905) are all greater than 0.9. These central cities have a good coupling degree with coastal cities, and agricultural informatization and agricultural economic development have a great degree of mutual influence and restriction. Compared with the central city, the coupling degree of surrounding cities is poor, and the degree of influence restriction is small, such as the case in Dongying City (0.241) and Rizhao City (0.385).

**Table 7. Level of coupling degree and coordination degree in Shandong Province.**

| Year | Coupling | Coordination index | Coordination degree | Coordination level | Coordination degree |
|------|----------|--------------------|--------------------|--------------------|---------------------|
| 2011 | 0.145 | 0.179 | 0.161 | 2 | A serious imbalance between |
| 2012 | 0.423 | 0.238 | 0.317 | 4 | Mild disorder |
| 2013 | 0.638 | 0.363 | 0.481 | 5 | On the verge of disorder |
| 2014 | 0.755 | 0.397 | 0.547 | 6 | Barely coordinated |
| 2015 | 0.793 | 0.419 | 0.577 | 6 | Barely coordinated |
| 2016 | 0.660 | 0.462 | 0.553 | 6 | Barely coordinated |
| 2017 | 0.710 | 0.546 | 0.622 | 7 | Mild coordination |
| 2018 | 0.482 | 0.603 | 0.539 | 6 | Barely coordinated |
| 2019 | 0.701 | 0.803 | 0.750 | 8 | Moderate coordination |

Data source: Calculated using coupling degree and coordination degree model.

**Table 8. Coupling degree and coordination degree of urban agricultural informatization and agricultural economic development in Shandong Province.**

| City | 2016 | | | | 2017 | | | |
|---|---|---|---|---|---|---|---|---|
| | Coupling | Coordination degree | Coordination level | Coordination degree | Coupling | Coordination degree | Coordination level | Coordination degree |
| Jinan | 0.965 | 0.850 | 9 | Good coordination | 0.952 | 0.843 | 9 | Good coordination |
| Qingdao | 0.977 | 0.922 | 10 | High-quality coordination | 0.986 | 0.941 | 10 | High-quality coordination |
| Zibo | 0.700 | 0.421 | 5 | On the verge of disorder | 0.591 | 0.400 | 5 | On the verge of disorder |
| Zaozhuang | 0.749 | 0.315 | 4 | Mild disorder | 0.607 | 0.244 | 3 | Moderate disorders |
| Dongying | 0.229 | 0.217 | 3 | Moderate disorders | 0.330 | 0.275 | 3 | Moderate disorders |
| Yantai | 0.938 | 0.718 | 8 | Moderate coordination | 0.938 | 0.739 | 8 | Moderate coordination |
| Weifang | 0.948 | 0.759 | 8 | Moderate coordination | 0.958 | 0.776 | 8 | Moderate coordination |
| Jining | 0.886 | 0.617 | 7 | Mild coordination | 0.897 | 0.624 | 7 | Mild coordination |
| Tai'an | 0.866 | 0.466 | 5 | On the verge of disorder | 0.878 | 0.483 | 5 | On the verge of disorder |
| Weihai | 0.648 | 0.429 | 5 | On the verge of disorder | 0.665 | 0.448 | 5 | On the verge of disorder |
| Rizhao | 0.289 | 0.197 | 2 | A serious imbalance between | 0.433 | 0.272 | 3 | Moderate disorders |
| Linyi | 0.849 | 0.581 | 6 | Barely coordinated | 0.883 | 0.616 | 7 | Mild coordination |
| Dezhou | 0.793 | 0.414 | 5 | On the verge of disorder | 0.801 | 0.427 | 5 | On the verge of disorder |
| Liaocheng | 0.822 | 0.399 | 4 | Mild disorder | 0.624 | 0.325 | 4 | Mild disorder |
| Binzhou | 0.778 | 0.402 | 5 | On the verge of disorder | 0.820 | 0.421 | 5 | On the verge of disorder |
| Heze | 0.494 | 0.290 | 3 | Moderate disorders | 0.469 | 0.308 | 4 | Mild disorder |
| City | 2018 | | | | 2019 | | | |
| | Coupling | Coordination degree | Coordination level | Coordination degree | Coupling | Coordination degree | Coordination level | Coordination degree |
| Jinan | 0.949 | 0.843 | 9 | Good coordination | 0.988 | 0.922 | 10 | High-quality coordination |
| Qingdao | 0.989 | 0.954 | 10 | High-quality coordination | 0.987 | 0.953 | 10 | High-quality coordination |
| Zibo | 0.592 | 0.397 | 4 | Mild disorder | 0.874 | 0.511 | 6 | Barely coordinated |
| Zaozhuang | 0.579 | 0.237 | 3 | Moderate disorders | 0.858 | 0.375 | 4 | Mild disorder |
| Dongying | 0.241 | 0.252 | 3 | Moderate disorders | 0.204 | 0.219 | 3 | Moderate disorders |
| Yantai | 0.936 | 0.740 | 8 | Moderate coordination | 0.962 | 0.746 | 8 | Moderate coordination |
| Weifang | 0.905 | 0.736 | 8 | Moderate coordination | 0.969 | 0.795 | 8 | Moderate coordination |
| Jining | 0.892 | 0.626 | 7 | Mild coordination | 0.899 | 0.646 | 7 | Mild coordination |
| Tai'an | 0.867 | 0.509 | 6 | Barely coordinated | 0.822 | 0.479 | 5 | On the verge of disorder |
| Weihai | 0.596 | 0.448 | 5 | On the verge of disorder | 0.410 | 0.331 | 4 | Mild disorder |
| Rizhao | 0.385 | 0.217 | 3 | Moderate disorders | 0.351 | 0.239 | 3 | Moderate disorders |
| Linyi | 0.703 | 0.495 | 5 | On the verge of disorder | 0.765 | 0.616 | 7 | Mild coordination |
| Dezhou | 0.821 | 0.444 | 5 | On the verge of disorder | 0.902 | 0.477 | 5 | On the verge of disorder |

*(Continued)*

**Table 8.** (Continued)

| Liaocheng | 0.864 | 0.419 | 5 | On the verge of disorder | 0.810 | 0.462 | 5 | On the verge of disorder |
|---|---|---|---|---|---|---|---|---|
| Binzhou | 0.749 | 0.386 | 4 | Mild disorder | 0.831 | 0.394 | 4 | Mild disorder |
| Heze | 0.655 | 0.407 | 5 | On the verge of disorder | 0.416 | 0.350 | 4 | Mild disorder |

Data source: Calculated using coupling degree and coordination degree model.

## Coordination analysis

As shown in Table 7, the coupling coordination degree between agricultural informatization and agricultural economic development in Shandong Province increases from 0.161 in 2011 to 0.622 in 2017, then decreases to 0.539 in 2018, and then increases to 0.75 in 2019. On the whole, the degree of coupling coordination between agricultural informatization and agricultural economic development in Shandong Province gradually becomes better, from a serious imbalance to moderate coordination. From 2011 to 2013, there is no coordination between agricultural informatization and agricultural economic development. Starting in 2014, agricultural informatization and agricultural economic development become gradually coordinated, and the coupling coordination reaches the highest level (0.750) in 2019. This is mainly because, in 2014, the Ministry of Agriculture of China issued the Notice on Carrying out the Pilot Work of Introducing Information into Villages and Households, which effectively improved farmers' ability to obtain information.

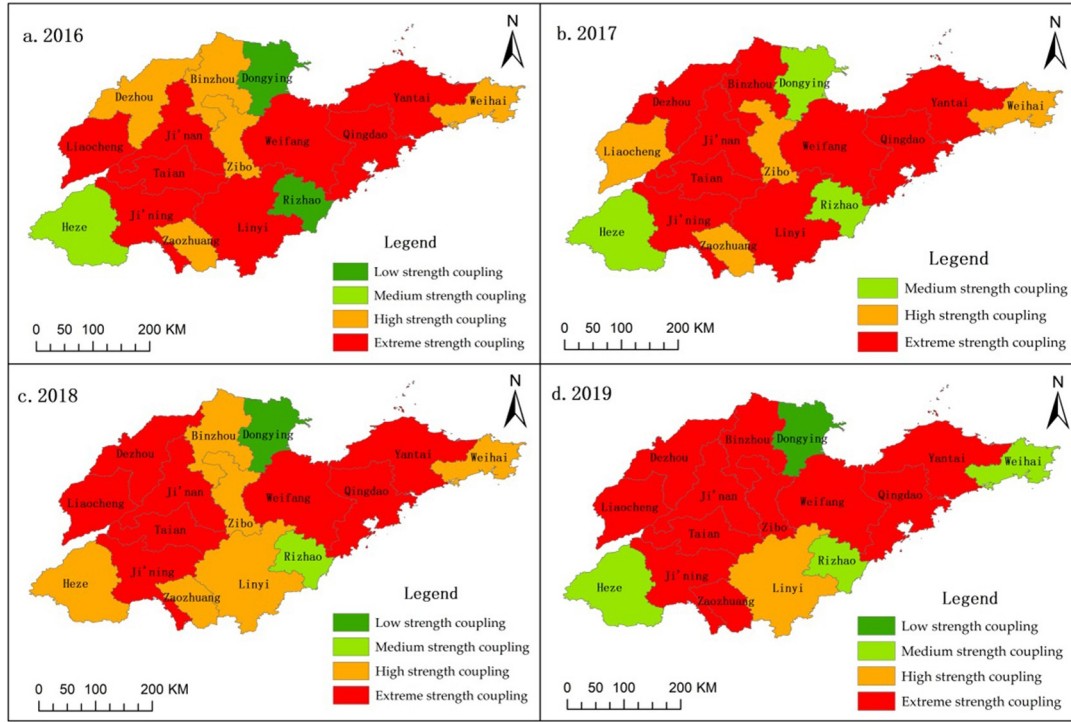

**Fig 2. Spatial distribution pattern of coupling degree between urban agricultural informatization and agricultural economic development in Shandong Province, 2016 to 2019.**

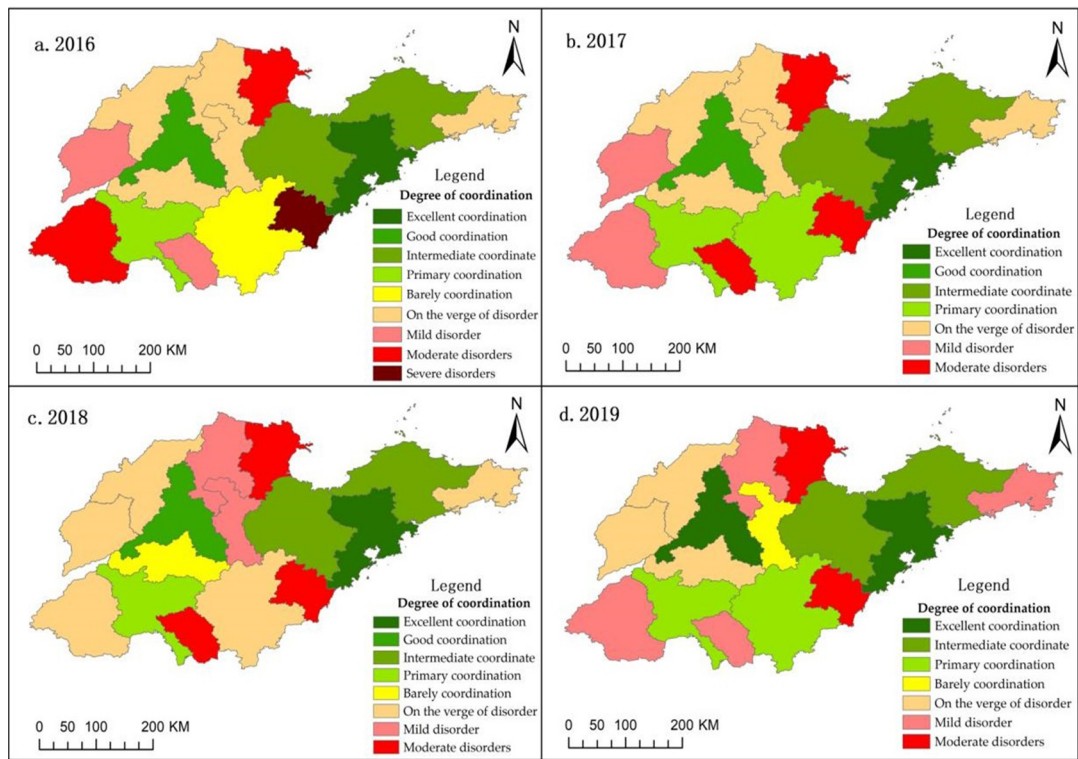

**Fig 3. Spatial distribution pattern of coordination degree between urban agricultural informatization and agricultural economic development in Shandong Province, 2016 to 2019.**

In 2015, the country actively promoted the "Internet +" modern agriculture action, vigorously developed smart agriculture, and further promoted the construction of agricultural IoT, agricultural e-commerce, and big data. To a certain extent, the development of agricultural informatization in Shandong Province in 2014 and 2015 laid a foundation for the subsequent agricultural informatization construction and also promoted the coordinated development between the two in the following years. With the rapid development of smart agriculture and digital agriculture in Shandong Province, the construction of agricultural informatization is constantly improved. The level of coordinated development between agricultural informatization and agricultural economy has gone through the transition period of having been barely coordinated and then having been successfully developed toward the direction of moderate coordination, good coordination, and high-quality coordination. As shown in Table 8, the numerical span of the coupling coordination degree between agricultural informatization and agricultural economic development in Shandong Province from 2016 to 2019 is 0.197 to 0.954, and the coordination degree is mainly moderate disorder and moderate coordination. Although the overall situation of agricultural informatization and agricultural economy in Shandong province has tended to develop benignly in the past four years, there is a large gap within the province. The degree of coordination in Jinan and Qingdao is relatively high, whereas that in Dongying and Lizhao is generally low.

The ArcGIS platform was further used to conduct spatial visualization processing of the coordination degree between agricultural informatization and agricultural economic development in Shandong Province from 2016 to 2019, as shown in Fig 3. As shown in the figure, Qingdao has the best coordination degree from 2016 to 2019, and Weihai and Weifang, adjacent to Qingdao, also have a relatively good coordination degree between agricultural

informatization and agricultural economic development. This clearly shows that the development of agricultural informatization and agricultural economy in Qingdao plays a leading role in the surrounding cities. In addition, the degree of coordination in the spatial pattern shows that the degree of coordination in the eastern region is stronger than that in the western region. The coordination degree of Heze City, Liaocheng City, and Dongying City is at the bottom of the whole province, which needs to be improved. Therefore, it is necessary to give full play to the leading role of key cities, further optimize the measures taken by Qingdao and Jinan, and improve the coordination of agricultural informatization and agricultural economic development in surrounding cities.

## Discussion and enlightenment

### Discussion

Following the prosperity and development of the Internet, big data, and other new-generation information technology, agricultural informatization and agricultural economic development have gradually shown a development trend of mutual promotion and mutual integration. This growth effectively promotes the prosperity and development of the digital economy in various regions. With the continuous improvement of agricultural informatization construction, the agricultural economy is also developing in a more prosperous direction. The coordinated development of the two can be discussed from the following aspects.

First, the study shows that agricultural information talents are insufficient. Although Shandong Province has an abundant labor force, the educational level of agricultural informatization talents is generally low, and the number of high-level agricultural informatization talents is insufficient. Moreover, some agricultural informatization talent has a lack of willingness to learn independently and a lack of enthusiasm in mastering and understanding new knowledge. Agricultural informatization needs combined talent who have agriculture-related knowledge and computer and other information technologies [28]. Otherwise, even with the guidance of information technology personnel, computers cannot be effectively used for agricultural informatization software development and agricultural data analysis [63].

Second, research also has found that agricultural information infrastructure construction lags. At present, Shandong Province's agricultural information network platform construction and operation mechanism is not perfect, and the relevant information service facilities cannot well meet the needs of farmers [64]. Although local government departments at all levels have set up their own agricultural information service platform, some of the villages and towns have no corresponding agricultural informatization service platform. The farmers may have access only to television media, village notices of the present situation of agricultural development, and related preferential policies. This may cause the farmer to accept information too late, which can restrict the development level of the agricultural economy.

Third, funding for agricultural informatization is insufficient. The cultivation of compound talents in agricultural informatization, the infrastructure construction of the IoT, the improvement of the agricultural information service platform, big data analysis, and the demonstration and application of agricultural information technology all need long-term and lasting financial guarantee [62]. The relevant government departments and farmers have not provided enough financial support for the construction of agricultural informatization. Therefore, the existing funds cannot fully train compound talents and improve agricultural information facilities, which restricts the innovation and development of information technology in the field of agriculture.

In addition, farmers have a weak awareness of informatization in the development of agricultural informatization and the agricultural economy. In recent years, with the deepening of

the integration of the new generation of information technology and the agricultural field, the problem of farmers' weak awareness of informatization in the development of agricultural informatization and the agricultural economy has become increasingly obvious. Only when farmers truly realize the importance of information technology can further integration of agriculture and information technology be achieved. At present, the ways to improve farmers' information consciousness mainly include information technology training courses and public lectures. The challenge of the current work, however, is how to arouse people's enthusiasm for learning. In the process of agricultural informatization construction, both the government and farmers should play their respective roles. The government should ensure that farmers have timely access to agricultural information in the agricultural information market and ensure that the relevant agricultural policies formulated by the government can be fully implemented. The construction of China's agricultural information market is still not perfect, since there are no policies to guarantee farmers' timely access to relevant agricultural information. Farmers should take the initiative to understand information related to agricultural informatization and actively participate in the construction of agricultural informatization [65]. To date, however, the enthusiasm of farmers is not high.

## This study has some important practical implications

**Accelerate the training of interdisciplinary talents.** It is important to train compound talents in agricultural informatization to better develop intelligent agriculture and promote agricultural informatization construction. The Shandong Provincial government can cooperate with provincial universities to jointly create agricultural informatization talents and encourage college graduates to return to their hometowns to start businesses. Agricultural and forestry colleges and universities in Shandong Province should be encouraged to set up courses related to agricultural informatization to cultivate college students' agricultural informatization construction ability. Colleges and universities can play a role in cultivating a strong learning ability and interest in the agricultural informatization talent team and increasing capital investment in specialized technical training and talent training in rural areas with professional talents to drive farmers to improve the cognition of agricultural informatization and action.

**Improve the infrastructure.** We will improve the development of agricultural information-based equipment in rural areas, continuously improve the information-based level of infrastructure, such as agricultural machinery and equipment, carry out processing and marketing of agricultural products, accelerate the implementation of the "Internet +" modern agriculture process, and promote pilot demonstrations of agricultural informatization. The regional pilot project for the agricultural IoT, agricultural e-commerce demonstration project, agricultural data investigation and analysis system construction project, agricultural information economy demonstration zone construction project, and information access to villages and households project will be fully implemented. We will further improve the agricultural information service system so that the information service system involves all aspects of agriculture and ensures the timely, accurate, and complete agricultural information service.

**Improve agricultural-informatization-related policies and regulations.** The government has plans to further promote the "Internet + agriculture," encourage the construction of agricultural informatization, and promote the development of the agricultural economy. Therefore, it is particularly important to improve the relevant policies and regulations of agricultural informatization construction: first, establish and improve agricultural-informatization-related technical specifications; second, improve the talent training mechanism and market competition mechanism, attract high-level talent, and cultivate compound talents; and, third, establish and perfect policies to encourage research and development (R&D) and

innovation and encourage software innovation, information technology breakthroughs, and related agricultural informatization product R&D.

**Improve farmers' awareness of agricultural informatization construction.**   Firstly, improve publicity so that farmers understand the status of agricultural informatization. In addition, focus on the publicity of the benefits brought by the construction of agricultural informatization so that farmers deeply understand the benefits of participating in the construction of agricultural informatization. This way, more people can support and participate in the construction of agricultural informatization. Second, strengthen the study of computing. At present, China's development goal is to build modern power and fully apply the new generation of information technology to all fields. Therefore, actively learning computer technology has also laid a solid theoretical basis for the development of "Internet + agriculture." Third, increase the information training of leading personnel. The degree to which the top leaders attach importance to informatization directly affects the actions of the subordinate employees on the construction of agricultural informatization.

**Strengthen the integration of information technology and agriculture with regional characteristics.**   Adjust measures to local conditions, give full play to advantages, and rely on information to promote characteristic agricultural information development. Agricultural production relies on advanced 5G technology, cloud computing, and big data to create "smart agriculture" and realize on-demand production. Sales of agricultural products through the "Internet + rural e-commerce" further expand the sales channels of agricultural products. Through information technology, an intelligent product quality supervision system can be created for agricultural product quality and safety management.The transportation of agricultural products is the last step in rural transportation before reaching farmers with the help of smart logistic technology.

## Conclusions and limitations

### Conclusions

In this paper, Shandong Province is selected as the research object, and through the construction of the coupling degree evaluation model and coordination degree evaluation model, the coordination relationship between agricultural informatization and agricultural economic development in Shandong Province is identified and analyzed. Through in-depth analysis of the full text, the following research conclusions can be drawn. First, the level of agricultural informatization plays a vital role in coordinating the development of agricultural informatization and the agricultural economy. Thus, speeding up the construction of agricultural informatization has become a general trend. Second, on the whole, agricultural informatization and agricultural economic development in Shandong Province are highly interdependent and mutually restricted, and the degree of coordination between agricultural informatization and agricultural economic development shows a significant trend of increase, from 0.161 in 2011 to 0.75 in 2019. Third, from the regional point of view, from 2016 to 2019, highly coupled cities are the main urban types in Shandong Province, with Jinan (0.965) and Qingdao (0.977) having the largest impact, while Dongying (0.241) and Rizhao (0.385) having a smaller impact. Although the degree of coupling coordination among cities in the province tends to develop benignly, there is a large gap between cities, and the degree of coordination between cities in the eastern region is stronger than that in the western region. Fourth, although there are still some restrictive factors in the construction of agricultural informatization in Shandong Province, the policies and infrastructure construction related to agricultural informatization are improved. With the continuous innovation and development of agricultural informatization

in Shandong Province, agricultural output value and farmers' income level are improved, which greatly promotes the development of the agricultural economy.

## Limitations

Although this paper includes a comprehensive study on agricultural informatization and agricultural economic development in Shandong Province, it has some limitations. It tries to expand the scope of independent variables as much as possible; however, because of to the lack of some data, quantifiable data are given priority in the selection of evaluation indicators, and the impact of population psychological factors and human subjective initiative on agricultural informatization and agricultural economy is ignored.

## Supporting information

**S1 Data.**
(XLSX)

**S2 Data.**
(XLSX)

## Acknowledgments

We are indebted to the anonymous reviewers and editors.

## Author Contributions

**Conceptualization:** Yu Sun.

**Data curation:** Yu Sun.

**Formal analysis:** Yu Sun.

**Methodology:** Yu Sun, Mingquan Li.

**Writing – original draft:** Yu Sun, Zhe Zhao.

**Writing – review & editing:** Yu Sun.

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
