## [Decision Letter · Decision Letter 0]

9 May 2022

PONE-D-22-09771Coordination of agricultural informatization and agricultural economy development: A panel data analysis from Shandong Province, ChinaPLOS ONE

Dear Dr. Li,

Thank you for submitting your manuscript to PLOS ONE. After careful consideration, we feel that it has merit but does not fully meet PLOS ONE’s publication criteria as it currently stands. Therefore, we invite you to submit a revised version of the manuscript that addresses the points raised during the review process.

We look forward to receiving your revised manuscript.

Kind regards,

László Vasa, PhD

Academic Editor

PLOS ONE

Journal Requirements:

"This research was funded by Shandong Province Modern Agricultural Industrial Technology System (SDAIT-01-13) and the Project of Shandong Provincial Department of Education (No: M2020111) and Qingdao Social Science Planning Project: Research on the Interactive Relationship between Agricultural Informatization and Agricultural Economic Growth in Qingdao under the Background of Digital Rural Strategy（QDSKL2101221）and Qingdao Double Hundred Research Project: Research on the Coupling Mechanism and Guiding Policies of Digital Investment and Agricultural Modernization in Qingdao (2021-B-15) and Soft Science Project of Science and Technology Department of Shandong Province(2021RKY07139) ."

"N/A"

Reviewers' comments:

Reviewer's Responses to Questions

**Comments to the Author**

1. Is the manuscript technically sound, and do the data support the conclusions?

Reviewer #1: Yes

Reviewer #2: Yes

2. Has the statistical analysis been performed appropriately and rigorously? 

Reviewer #1: Yes

Reviewer #2: Yes

3. Have the authors made all data underlying the findings in their manuscript fully available?

Reviewer #1: Yes

Reviewer #2: Yes

4. Is the manuscript presented in an intelligible fashion and written in standard English?

Reviewer #1: Yes

Reviewer #2: Yes

5. Review Comments to the Author

Reviewer #1: The topic of the paper is relevant and very interesting. Authors used both national and international literature sources and cited them correctly. Most part of them are from the last 10 years. I suggest to improve literature source with international units. I agree that the continuous development of the world economy, science, and technology, the era of intelligence and information is upon us, which is true all sectors, such as the agriculture. Authors chose a major agricultural province in China, Shandong Province which has been ranked first in China for many years in terms of gross agricultural product and the import and export of agricultural products. This paper uses evaluation index data of Shandong Province from 2011 to 2019. Authors used different – relevant – methods in their research work. They presented their results and introduced them in a well-structured form using tables and figures. I appreciate their conclusions and agree those limitations which they wrote at the end of their research work.

Reviewer #2: Comments and Suggestions for Authors

Thank you for the opportunity to read the study paper. Topic is very interesting and topical. Topicality is obviously supported by the fact of agricultural informatization and its strong relation with agricultural economy development.

Title: The title is a little too long, but acceptable to me.

Abstract and introduction is well-structured and clear at the present form but the relevance and significance and novelty value of the entropy value method and coupling coordination degree analysis are missing. It should be more focused on the importance of the topic or main issue and gaps more.

The literature review. The literature review section (combined from introduction and overview of the study area) is good and well balanced.

Please correct or explain the followings:

A reference should be made to the table 2.1. In addition, all tables require resources, even if they are reconstructed or modified. What is the source of table 3.1?

Studying and interpret of table 3.2 is very difficult in the form of simple data, it is hard for humans, it is not communicate well, so a graphical representation of the data is necessary to draw further conclusions. A graph figure should be done it.

In the line 155 and 156: instead of capital ‘I’ use ‘i’

please harmonize ith; jth or i-th; j-th or JTH (My suggestion: ith or jth)

In the line 176: where n is the number of years

In the line 177: please explain “information entropy” or make a reference for it

In the line 185: n and k is the same, so use n instead

In the line 190: Ui where Ui represents the indicator in the i-th year (if I’m right or make it clear)

In table 4.1 I recommend the same or similar expressions if possible on both side (minus and plus). The ‘primary coordination’ is unclear (I suggest ‘mild coordination’) … moderate coordination, good or serious coordination, high-quality coordination

Discussion and enlightenment – This part is very good, but most of the findings seem to me to be a strategic program for the government or municipalities, not implications. Based on this, I suggest focusing more where possible on specific, short-term actions rather than a long-term focus or concept. (Although trainings and awareness raising are definitely good options.) If possible, also formulate generalized results that are not concerning to Shandong Province only.

Research findings (Conclusions and limitations) – These section is appropriate.

The paper has very good potential, but the authors need to make a minor revision at this stage. I’m very glad to have the opportunity to read your work.

6. PLOS authors have the option to publish the peer review history of their article (what does this mean?). If published, this will include your full peer review and any attached files.

Reviewer #1: No

Reviewer #2: **Yes: **Dr. Gyenge Balázs

---

## [Author Response · Author response to Decision Letter 0]

1 Jul 2022

I suggest to improve literature source with international units.

Reviewer #1: The topic of the paper is relevant and very interesting. Authors used both national and international literature sources and cited them correctly. Most part of them are from the last 10 years. I suggest to improve literature source with international units. I agree that the continuous development of the world economy, science, and technology, the era of intelligence and information is upon us, which is true all sectors, such as the agriculture. Authors chose a major agricultural province in China, Shandong Province which has been ranked first in China for many years in terms of gross agricultural product and the import and export of agricultural products. This paper uses evaluation index data of Shandong Province from 2011 to 2019. Authors used different – relevant – methods in their research work. They presented their results and introduced them in a well-structured form using tables and figures. I appreciate their conclusions and agree those limitations which they wrote at the end of their research work.

Thank you very much for your valuable suggestion. Based on your suggestion, I have added 12 international literature sources.

Reviewer #2: Comments and Suggestions for Authors

Thank you for the opportunity to read the study paper. Topic is very interesting and topical. Topicality is obviously supported by the fact of agricultural informatization and its strong relation with agricultural economy development.

Title: The title is a little too long, but acceptable to me.

Thank you very much for your valuable suggestion.

1. Abstract and introduction is well-structured and clear at the present form but the relevance and significance and novelty value of the entropy value method and coupling coordination degree analysis are missing. It should be more focused on the importance of the topic or main issue and gaps more.

The literature review. The literature review section (combined from introduction and overview of the study area) is good and well balanced.

The relevance between them was added to section 4.2, and innovation and research significance were added to section 4.1.

2. Please correct or explain the followings:

A reference should be made to the table 2.1. In addition, all tables require resources, even if they are reconstructed or modified. What is the source of table 3.1?

I added the source of literature to all the tables, thank you.

3. Studying and interpret of table 3.2 is very difficult in the form of simple data, it is hard for humans, it is not communicate well, so a graphical representation of the data is necessary to draw further conclusions. A graph figure should be done it.

I have added another chart as Figure 3.1 in the corresponding position. The table and the chart can be seen more clearly together. Thank you.

4. In the line 155 and 156: instead of capital ‘I’ use ‘i’

please harmonize ith; jth or i-th; j-th or JTH (My suggestion: ith or jth)

In the line 176: where n is the number of years

In the line 177: please explain “information entropy” or make a reference for it

In the line 185: n and k is the same, so use n instead

In the line 190: Ui where Ui represents the indicator in the i-th year (if I’m right or make it clear)

In table 4.1 I recommend the same or similar expressions if possible on both side (minus and plus). The ‘primary coordination’ is unclear (I suggest ‘mild coordination’) … moderate coordination, good or serious coordination, high-quality coordination

I sincerely apologize for the mistake. I have made some modifications to solve these problems.

Because these are coordination levels, I looked up some literature and found that they are not plus or minus signs. Thank you very much for your valuable suggestion.

5. Discussion and enlightenment – This part is very good, but most of the findings seem to me to be a strategic program for the government or municipalities, not implications. Based on this, I suggest focusing more where possible on specific, short-term actions rather than a long-term focus or concept. (Although trainings and awareness raising are definitely good options.) If possible, also formulate generalized results that are not concerning to Shandong Province only.

I have deleted “Give full play to the principal role of the government” and added new content, “Strengthen the integration of information technology and agriculture with regional characteristics”.

Research findings (Conclusions and limitations) – These section is appropriate.

The paper has very good potential, but the authors need to make a minor revision at this stage. I’m very glad to have the opportunity to read your work.

Thank you very much.

---

## [Editor Report · Decision Letter 1]

12 Jul 2022

PONE-D-22-09771R1Coordination of agricultural informatization and agricultural economy development: A panel data analysis from Shandong Province, ChinaPLOS ONE

Dear Dr. Li,

Thank you for submitting your manuscript to PLOS ONE. After careful consideration, we feel that it has merit but does not fully meet PLOS ONE’s publication criteria as it currently stands. Therefore, we invite you to submit a revised version of the manuscript that addresses the points raised during the review process. In the submitted revised version I can't see the changes. You attached both files but in one of them, the changes should be indicated with red. Please prepare it accordingly and submit it again.

Please submit your revised manuscript by 31.07.2022 If you will need more time than this to complete your revisions, please reply to this message or contact the journal office at plosone@plos.org. Please include the following items when submitting your revised manuscript:A rebuttal letter that responds to each point raised by the academic editor and reviewer(s). You should upload this letter as a separate file labeled 'Response to Reviewers'.A marked-up copy of your manuscript that highlights changes made to the original version. You should upload this as a separate file labeled 'Revised Manuscript with Track Changes'.An unmarked version of your revised paper without tracked changes. You should upload this as a separate file labeled 'Manuscript'.If applicable, we recommend that you deposit your laboratory protocols in protocols.io to enhance the reproducibility of your results. Protocols.io assigns your protocol its own identifier (DOI) so that it can be cited independently in the future. For instructions see: https://journals.plos.org/plosone/s/submission-guidelines#loc-laboratory-protocols. Additionally, PLOS ONE offers an option for publishing peer-reviewed Lab Protocol articles, which describe protocols hosted on protocols.io. Read more information on sharing protocols at https://plos.org/protocols?utm_medium=editorial-email&utm_source=authorletters&utm_campaign=protocols.

We look forward to receiving your revised manuscript.

Kind regards,

László Vasa, PhD

Academic Editor

PLOS ONE
---

## [Author Response · Author response to Decision Letter 1]

17 Jul 2022

Reviewer #1: The topic of the paper is relevant and very interesting. Authors used both national and international literature sources and cited them correctly. Most part of them are from the last 10 years. I suggest to improve literature source with international units. I agree that the continuous development of the world economy, science, and technology, the era of intelligence and information is upon us, which is true all sectors, such as the agriculture. Authors chose a major agricultural province in China, Shandong Province which has been ranked first in China for many years in terms of gross agricultural product and the import and export of agricultural products. This paper uses evaluation index data of Shandong Province from 2011 to 2019. Authors used different – relevant – methods in their research work. They presented their results and introduced them in a well-structured form using tables and figures. I appreciate their conclusions and agree those limitations which they wrote at the end of their research work.

Thank you very much for your valuable suggestion. Based on your suggestion, I have added 12 international literature sources.

Reviewer #2: Comments and Suggestions for Authors

Thank you for the opportunity to read the study paper. Topic is very interesting and topical. Topicality is obviously supported by the fact of agricultural informatization and its strong relation with agricultural economy development.

Title: The title is a little too long, but acceptable to me.

Thank you very much for your valuable suggestion.

1. Abstract and introduction is well-structured and clear at the present form but the relevance and significance and novelty value of the entropy value method and coupling coordination degree analysis are missing. It should be more focused on the importance of the topic or main issue and gaps more.

The literature review. The literature review section (combined from introduction and overview of the study area) is good and well balanced.

The relevance between them was added to section 4.2, and innovation and research significance were added to section 4.1.

2. Please correct or explain the followings:

A reference should be made to the table 2.1. In addition, all tables require resources, even if they are reconstructed or modified. What is the source of table 3.1?

I added the source of literature to all the tables, thank you.

3. Studying and interpret of table 3.2 is very difficult in the form of simple data, it is hard for humans, it is not communicate well, so a graphical representation of the data is necessary to draw further conclusions. A graph figure should be done it.

I have added another chart as Figure 3.1 in the corresponding position. The table and the chart can be seen more clearly together. Thank you.

4. In the line 155 and 156: instead of capital ‘I’ use ‘i’

please harmonize ith; jth or i-th; j-th or JTH (My suggestion: ith or jth)

In the line 176: where n is the number of years

In the line 177: please explain “information entropy” or make a reference for it

In the line 185: n and k is the same, so use n instead

In the line 190: Ui where Ui represents the indicator in the i-th year (if I’m right or make it clear)

In table 4.1 I recommend the same or similar expressions if possible on both side (minus and plus). The ‘primary coordination’ is unclear (I suggest ‘mild coordination’) … moderate coordination, good or serious coordination, high-quality coordination

I sincerely apologize for the mistake. I have made some modifications to solve these problems.

Because these are coordination levels, I looked up some literature and found that they are not plus or minus signs. Thank you very much for your valuable suggestion.

5. Discussion and enlightenment – This part is very good, but most of the findings seem to me to be a strategic program for the government or municipalities, not implications. Based on this, I suggest focusing more where possible on specific, short-term actions rather than a long-term focus or concept. (Although trainings and awareness raising are definitely good options.) If possible, also formulate generalized results that are not concerning to Shandong Province only.

I have deleted “Give full play to the principal role of the government” and added new content, “Strengthen the integration of information technology and agriculture with regional characteristics”.

Research findings (Conclusions and limitations) – These section is appropriate.

The paper has very good potential, but the authors need to make a minor revision at this stage. I’m very glad to have the opportunity to read your work.

Thank you very much.

---

## [Editor Report · Decision Letter 2]

3 Aug 2022

Coordination of agricultural informatization and agricultural economy development: A panel data analysis from Shandong Province, China

PONE-D-22-09771R2

Dear Dr. Li,

We’re pleased to inform you that your manuscript has been judged scientifically suitable for publication and will be formally accepted for publication once it meets all outstanding technical requirements.

Kind regards,

László Vasa, PhD

Academic Editor

PLOS ONE
---

## [Editor Report · Acceptance letter]

16 Aug 2022

PONE-D-22-09771R2 

Coordination of agricultural informatization and agricultural economy development: A panel data analysis from Shandong Province, China 

Dear Dr. Li:

I'm pleased to inform you that your manuscript has been deemed suitable for publication in PLOS ONE. Congratulations! Your manuscript is now with our production department. 

Kind regards, 

on behalf of

Prof. Dr. László Vasa 

Academic Editor

PLOS ONE